# Real-Time Massive Vector Field Data Processing in Edge Computing

**DOI:** 10.3390/s19112602

**Published:** 2019-06-07

**Authors:** Kun Zheng, Kang Zheng, Falin Fang, Hong Yao, Yunlei Yi, Deze Zeng

**Affiliations:** 1School of Geography and Information Engineering, China University of Geosciences, Wuhan 430074, China; 2Wuhan Zhaotu Technology Co. Ltd., Wuhan 430074, China; cugwhlin2014@126.com (F.F.); yiyunlei@zmap-tech.com (Y.Y.); 3School of Computer Science, China University of Geosciences, Wuhan 430074, China; yaohong@cug.edu.cn (H.Y.); deze@cug.edu.cn (D.Z.)

**Keywords:** edge computing, massive vector field data, fluidization, high-performance, framework

## Abstract

The spread of the sensors and industrial systems has fostered widespread real-time data processing applications. Massive vector field data (MVFD) are generated by vast distributed sensors and are characterized by high distribution, high velocity, and high volume. As a result, computing such kind of data on centralized cloud faces unprecedented challenges, especially on the processing delay due to the distance between the data source and the cloud. Taking advantages of data source proximity and vast distribution, edge computing is ideal for timely computing on MVFD. Therefore, we are motivated to propose an edge computing based MVFD processing framework. In particular, we notice that the high volume feature of MVFD results in high data transmission delay. To solve this problem, we invent Data Fluidization Schedule (DFS) in our framework to reduce the data block volume and the latency on Input/Output (I/O). We evaluated the efficiency of our framework in a practical application on massive wind field data processing for cyclone recognition. The high efficiency our framework was verified by the fact that it significantly outperformed classical big data processing frameworks Spark and MapReduce.

## 1. Introduction

The space occupied by fluid motion is called the flow field. As time goes on, the fluid presents different states in the space. These states can be collected by sensors and expressed as vector field data. With the development of various industrial applications, vector field data play an increasingly important role in various fields [1,2]. For example, in the meteorological field, wind field data can be used to analyze the wind field structure and conduct typhoon warning. Ocean current data can be used to analyze the impact of ocean current on climate [3,4]. On the other hand, with the continuous expansion and growth of data, the real-time application of massive vector field data (MVFD) is gradually facing various challenges. MVFD can be regarded as a kind of big data characterized by volume, velocity, variety and value. Beyond that, MVFD has its own extraordinary characteristics, such as distribution, real-time, high-volume, spatiality.

Distribution: MVFD is generated by sensors with vast distribution, and is continuously generated with the dynamics of the flow field state [5,6]. Therefore, processing MVFD at the centralized cloud may experience high latency due to the data transmission delay between the data source and the cloud.

Real-time: If the generated MVFD is not processed in time, the value of data will be gradually reduced. For example, the analysis of wind field and ocean current data should be timely processed for natural disaster prediction to send typhoon and tsunami warning information in advance to reduce loss [7,8,9].

High-Volume: The vector field data are used to describe complex fluid states [9,10]. In addition to time and space dimensions, a series of subsidiary features are usually required, e.g., direction, velocity, trend and so on. Large spatial range and multi-dimensional characteristics lead to large volume of single data. Existing analysis of the vector field data only focuses on a single or a few dimensions [1,6,11].

Spatiality: MVFD has strong spatial correlation in calculation. For example, Yuan et al. [12] proposed a vector field segmentation method based on Clifford-FFT and used a clustering algorithm to partition space. It is necessary to repeatedly use the adjacent grid points in iterative computation.

Considering the above characteristics, MVFD processing faces unprecedented challenges. Obviously, it is not efficient and economic to transmit bulk MVFD to the centralized cloud due to the long transmission delay. Nane Kratzke summarized much cloud-native research as well as its concepts and characteristics. Interested readers can refer to the work in [13] for more details. To solve the bottleneck of “long transmission delay” in cloud computing, researchers further proposed the solution of edge computing [14]. Compared with cloud computing, edge computing can explore the geo-distributed resources at the network edge, therefore become an ideal platform to host MVFD applications. Due to the inherent differences between edge computing and cloud computing, traditional cloud computing oriented big data processing frameworks such as MapReduce and Spark are not suitable for edge computing empowered MVFD processing any more. For example, if the data were divided into fine-scale data block and transmit to different edge servers randomly, considering the spatial correlation of data, there would be a lot of communication overhead between nodes in the calculation process. We need a fine-grain data transmission and computing scheduling mechanism. Targeting at the above challenges, in this paper, we are motivated to investigate how to efficiently use the edge computing resources to process MVFD. The main contributions of this paper are as follows:We propose an edge computing empowered MVFD processing framework. To our best knowledge, we are the first to introduce edge computing for MVFD processing in the literature.We invent Data Fluidization Schedule (DFS) strategy, aiming at fine-scale data partitioning, dataflow transmission and computing scheduling so as to ensure the real-time MVFD processing.We practically developed and implemented our framework with DFS. Real MVFD data-driven experiment results show that it is superior to both Spark and MapReduce.

The rest of paper is organized as follows: Section 2 introduces the related studies on massive vector data processing and edge computing. Section 3 describes the design of our edge computing empowered MVFD processing framework, and introduces a typical application scenario. Section 4 presents data fluidization scheduling strategy. Section 5 reports our trace-driven performance evaluations. Section 6 concludes this paper.

## 2. Related Work

In this section, we first discuss some previous studies on MVFD processing acceleration as well as some existing studies on applying edge computing for big data processing.

### 2.1. MVFD Processing

The vector field data are usually represented as a grid based discrete sample [15,16,17,18]. The generated vector field contains the information of the time domain. For timely MVFD processing, researchers have developed various MVFD processing acceleration from different aspects. Existing studies mainly focus on cloud computing based MFVD processing [19,20,21]. Christoph et al. [5] summarized them into five categories, namely parallelize-over-blocks (POB), parallelize-over-seeds (POS), adaptive load balancing, hybrid parallelism, NS extended memory hierarchies, where the latter three strategies are actually extended from the former two, as illustrated in Figure 1. Interested readers may refer to the work in [5] for more details. Here, we introduce the main strategies related to our work as follows.

POB: Each server preloads only part of the data and processes only the data blocks stored on it, as shown in Figure 1a. All results related to its own data block are output once all iterations are completed. The final result is stitched to the results of each server. When the calculation involves data in other servers, it need to retrieve the data from other nodes. Sujudi et al. [22] divided the blocks into several sub-blocks and assigned each processor a sub-block. Such method suffers high data communication overhead during the processing.

POS: Each server preloads all the data, completes partial iterative processing, and finally output the results, as shown in Figure 1b. Pugmire et al. [23,24] analyzed the POS performance and showed that, compared with POB, the advantage of POS is to reduce the data communication. However, it also leads to a lot of data redundancy, and the communication overhead is also high in data preparation.

Adaptive load balancing: In the process of data loading and calculation, the data block assignment and calculation are dynamically adjusted to achieve the equilibrium state of the calculation load on each node, in order to maximize the use of computing resources. For example, Boonthanome et al. [1] put forward two-way partitioning to achieve load balancing, to improve the efficiency of calculation.

Hybrid parallelism: It can be regarded as a combination of POS and POB. Some redundant data are loaded in each server to seek an equilibrium point in the data preparation and data processing. Camp et al. [25] investigated the performance efficiency of hybrid parallel implementation.

Extended memory hierarchies: Based on POS, the loaded data can be persisted in memory as long as possible, reducing data loading time and improving computing efficiency [5]. Camp et al. [26] also investigated the benefits of using an extended memory hierarchy in combination with the POS.

The timeliness of the application is not only dependent on the timeliness of the calculation, the loss of time from the data source to the computing environment and the final output of the result on the I/O also needs to be taken into consideration. On this basis, we are motivated to design an edge computing based framework for real-time MVFD processing, with an emphasis on data I/O optimization.

### 2.2. Edge Computing

It is widely recognized that cloud computing suffers from long processing latency as all the computations happen in the centralized cloud and all the data must be first collected into the cloud. This motivates the proposal of edge computing [27,28,29,30,31]. Edge computing allows computations to be performed at the edge of the network. “Edge” refers to any computing and network resources along the path between data sources and cloud data centers [32,33,34,35]. By such paradigm, edge computing becomes an alternative platform to host cloud computing tasks. For example, researchers use cloudlets to offload computing tasks from cloud computing. The results show that response time reduced by 80–200 milliseconds. Moreover, the energy consumption could also be reduced by 30–40% [36].

Thanks to such advantages, edge computing has been widely applied to a diversity of domains, e.g., video analytics, smart home, smart city and so on. In addition, with the consideration of inherent differences between cloud computing and edge computing, much effort has also been devoted to optimizing data processing at the edge computing. For example, Soraia et al. [37] proposed an edge computing based smart healthcare framework for resource management, which uses Petri net to optimize the allocation of resources. On this basis, the service time of various medical institutions can be significantly reduced. Ganesh et al. [38] used geo-distributed video analytics infrastructure to optimize the real-time video analytics. Mao et al. [39] adopted a Markov decision process approach to decide whether to execute a task locally at the mobile device or to offload it to an edge server.

## 3. Edge Computing Empowered MVFD Processing Framework

Vector field data are generated by sensors with wide distribution. Take wind field data as an example, where wind speed sensors are usually distributed in a large field. In this case, geo-distributed edge servers near these sensors are ideal to process, or partially process, the sensor data. Therefore, we propose an edge computing empowered MVFD framework, as shown in Figure 2. In our framework, edge computing is used as a complementary platform to cloud computing. The data processing tasks can be adaptively balanced according to the application requirements (e.g., processing delay). Both computation parallelism and data parallelism are supported in our framework. In computation parallelism, the edge computing can be used for some data preprocessing tasks, i.e., data partitioning, pre-calculation, and data splicing. These tasks are generally called as Data Fluidization Schedule (DFS) in MVFD processing, which has deep influence on the overall MVFD processing performance efficiency. We must carefully partition and reorganize the disordered flow data to reduce the data processing time. The detailed DFS design in our framework is elaborated in the next section. Here, let us proceed on the overview of our framework.

From the computation workflow and application development perspective, our framework has three layers, i.e., data layer, computation layer, and application layer (Figure 3). The data layer is mainly responsible for data partition and fluidization of the raw sensor data in the form of MVFD, which first pour into the data fluidization module, which make data partition and transmission scheduling decisions according to the system states (e.g., available resource capacities, data generation rate, and data size) and application requirements (e.g., application processing delay). Note that MVFD processing applications sometimes also require historical data query. To this end, distributed storage module is introduced to provide stable data storage. The computation layer takes the fluidized data from the data layer as input and process them according to the predefined application semantics. Usually, the process consists of different stages, which can be run in parallel. In the last stage, the data are directly output to the dataflow gate to ensure that the data are accurately exported to the real-time service set and the distributed storage module. The application layer provides two basic services, i.e., real-time update and historical search, based on which various MVFD applications can be developed. For real-time services, data buffer is introduced to ensure the stability and accuracy of data processing. For historical services, the data are structured and organized into the distributed storage module.

Next, we take meteorological application for cyclone recognition based on wind MVFD processing as an example to show the working process of our framework. Considering the structural characteristics of the cyclone as a regional cluster distribution, a square template with a scale of N×N is used for preliminary detection, where *N* is determined by the cyclone size and the resolution of vector field data. The distribution of wind direction in the template is used to determine whether it is a cyclone or not. For example, by inputting the vector field data (resolution is 0.25${}^{\circ }$× 0.25${}^{\circ }$) provided by European Meteorological Center in the range of 60–150${}^{\circ }$ longitude east, 10${}^{\circ }$ latitude south and 60${}^{\circ }$ latitude north. The range of each lattice point is then estimated to be about 14 km × 27 km. Assuming that the diameter of the cyclone is 100 km × 500 km, we have N=5,7,...,17 so that the template can cover the whole cyclone. Starting from the large-scale template, the wind direction data are matched in descending order until the small scale template matching is completed. To realize the above application, several computation stages are required and can be optimized by parallel acceleration, as shown in Figure 4. According to our proposed framework, the vector field data are divided into m×m sub-regions by space. Each sub-region expands the height and width of N/2 outward, so that the boundary area is completely covered. The partitioned data then need to be transmitted to the distributed edge servers for conversion and cyclone recognition. Thereafter, the cyclone data form a complete, unique dataset through the dataflow gate, and output the final results.

## 4. Data Fluidization Schedule

The core idea of DFS is using dataflow to process vector field data so as to reduce the time consumed in data preloading and output. In the workflow of dataflow process, each server can be regarded as an autonomous computational unit. Each server has an independent instruction stream and address space. The data shuffle between servers is realized through communication channels implemented as FIFO queues [40,41,42]. Based on this, to improve the computational efficiency of MVFD processing, we first carefully design the data partition pattern, as shown in Figure 5. The partitioned data are choreographed and transmitted in the form of dataflow, which are then iteratively processed by several stages defined in the MVFD processing algorithm. Finally, the results are output to the dataflow gate for verification of integrity and uniformity. Note that, in dataflow processing, parallel computing can be done at the same time of data transmission. The computation and communication should be carefully scheduled to achieve high MVFD processing efficiency.

### 4.1. Fluidization of Data

The purpose of data fluidization is to fluidize batch data into dataflow. In the process of data transmission, some data can be calculated to improve efficiency. Data fluidization mainly involve the following issues: data partition, data block coding, and dataflow transmission scheduling. For data partition, it requires that the spatial segmentation method must have the ability to reveal the characteristics of regional scope [43,44,45]. For spatial dimension, data are partitioned hierarchically. As shown in Figure 6, where *N* represents the hierarchy, the data are partitioned into sub-region and encoded according to its spatial position.

A large vector field data can be partitioned into (n−1+b) segments, where *n* and *b* are the total number of data dimension and the number of segments in spatial dimensions, respectively. Data segmentation needs to be set in sequence according to the weight of the dimension. Assuming that there are *m* different application scenarios, the weight $W_{i}$ of the dimension *i* is the sum of all application scene weights, i.e.,
(1)$$W_{i}=\sum_{j\in (1,m)}W_{ij}.$$

The high weight dimension should be segmented by the back, to ensure that the data with high weight dimensions are more complete in a short time.

First, the data are divided into data blocks sub−1−batch−1, sub−1−batch−2, and  sub−1−batch−t according to dimension1. Then, we encode these blocks in turn as code11,code12, and code1t. In the same way, the blocks are divided according to dimension2. For the data block sub−2−batch−j, it is encoded by combining code1j and code2j. Performing the same operation for each dimension, the final data encoding is made up of all the codes, i.e.,
(2)$$Code_{j}=\cup_{x\in (1,n)}Code_{xj}.$$

The number of outgoing partitions of the data stream should be as far as possible the same as the number of edge servers. This not only ensures that the edge resources are fully utilized, but also can reduce the communication cost. Based on this, assuming that the number of edge servers is *p*, *p* threads are created to submit the data blocks. Based on the segmentation pattern mentioned above, we use *p* threads to read data from the beginning of the data block sub−n−batch−1, sub−n−batch−2, and sub−n−batch−j in order and output the current data slices individually. Algorithm 1 provides pseudocode for the data partition and transmission scheduling.

**Algorithm 1** Partition and transmission of MVFD in sequence.**Require:** Raw Dataset, *D*;
Dimension sequence, *S*;Segmentation rules for the last dimension, *R*;Data Transfer Thread Pool, DP;
**Ensure:** Data stream consisting of data blocks;
1:For the first dimension $S_{1}$ in S, get all the data $D_{1}$ of the first value $S_{1}$[0] from D.2:For $S_{2}$, repeat the same work, and so on.;3:For the last dimension $S_{L}$, use *R* to partition the data and get the array $D_{S}$[];4:For each $D_{S}$[] generated, the loop submits each element to the thread pool DP;

### 4.2. Parallel Computing on Dataflow

The batch data are divided and processed as dataflow. However, for MVFD, dataflow formed by data partition are not independent. There exists certain correlation between the datasets. Therefore, parallel computing on dataflow also requires some ways to deal with such correlation. In this paper, we define the relationship between resource datasets and result datasets through single computation as parent and child datasets. There are two types of relationships between parent and child datasets: wide dependency type and narrow dependency type. With narrow dependency, all the data of a child dataset can be calculated by a single parent dataset, e.g., the extraction of points of the vector field data and the linear transformation of the data. With wide dependency, a child dataset is calculated from multiple parent datasets, e.g., data compression, line and surface extraction of vector field data. When the massive vector dataflow is transferred to the distributed edge computing platform, data may arrive at the intended edge server in disorder. In narrow dependency types, the order of dataset does not affect the results. However, in the case of wide dependent types, if the data involved in the processing are not yet transmitted, it will inevitably lead to errors. To tackle this issue, we divide parallel tasks into stages and execute them in stage by stage. Each stage is composed of all the calculations before the wide dependency. After the previous stage calculation is completed, the next stage can start the calculation to ensure data visibility between nodes. This means that the results of the previous stage may need to be shuffled between different edge servers. The last stage outputs results to the dataflow gate. The whole MVFD process can be divided into a number of different stages. The computation can be carried out during the data transmission. However, the child stage needs to wait until its parent dataset is completed. As shown in Figure 7, the wide dependency type in the calculation process includes the calculation from *A* to *C* and the calculation from *B* to *D* in the first two stages. Finally, Stage 3, as the last stage, outputs the results to the dataflow gate.

### 4.3. Dataflow Gate

In data fluidization procedure, the data from the same dimension are partitioned. The reversal of data transmission sequence may lead to incomplete data in the same batch. In addition, the data calculated at different edge servers may result in redundancy. To maintain data integrity and data uniqueness, the dataflow gate is designed to verify the data’s integrity and uniqueness. In fact, the result collection is dependent on multiple parent datasets. To improve the output efficiency, dynamic data buffer is adopted in the dataflow gate. It dynamically generates buffer for new dataset, and uses prefix code to delineate buffer data range. For example, when the latitude and longitude range is the final data partition basis, the dataset that encodes the same prefix keeps continuity on the plane space. Accordingly, in the data gate, these data are stored in the same buffer. When the data of the query range are completed, the whole buffer is exported to the customer node, as shown in Figure 8. As a result, there are two issues to be considered at this stage. The first issue is to maintain the consistency of data in the same dimension. The data with the same key value prefix is stored in the same continuous space. The second issue refers to the repeated data, which are covered by the same key value. The data block should be sent to the output data stream when the data block is judged as completed. Algorithm 2 provides pseudocode for the dataflow gate implementation based on the above principle.

**Algorithm 2** Dataflow gate.**Require:** Data consumer threads, Consumer;
Map of data buffers, Map[Buffer];Buffer size threshold, *T*;
**Ensure:** final result set,res;
  1:**while** Consumer.listening **do**  2:    **if**
Consumer.hasNewElement **then**  3:        element=consumer.newElement;  4:        key=getKeyPrefix(element);  5:        buffer=Map[Buffer].get(key);  6:        **if**
buffer.contains(element) **then**  7:            continue;  8:        **else**  9:            buffer+=element;10:        **end if**11:        **if**
buffer.length=*T*
**then**12:            res=buffer;13:            post(res);14:            buffer=null;15:        **end if**16:    **end if**17:**end while**

## 5. Experiment and Analysis

In this section, we report our experiment results by applying our framework for a practical meteorological application. Typical meteorological applications receive field data in size of terabytes per day, which need to be processed in time. However, usually it takes a long time to process such massive data. It is significant to improve the MVFD processing efficiency for such kinds of applications [46]. Therefore, we are motivated to use such applications to evaluate the efficiency of our framework.

### 5.1. Implementation and Settings

We applied our framework for cyclone recognition by processing meteorological wind field data. The experimental data are in NETCDF format, which includes six dimensions, i.e., longitude, latitude, time, elevation, uv vector (direction vector) and predictive data. Among them, the uv vector represents the wind speed and wind direction in the east–west direction and the north–south direction, respectively. The predictive data, based on the real-time data, predict the wind direction wind speed according to the time lapse. Two real datasets were used in our experiments The first dataset is wind field data in Guangdong, China, in May 2017. The size is 160 Gb, in range 593 × 393, five elevation segments and 1-h update frequency. The data are relatively discrete and the single data size is relatively small. The second wind field dataset is for 31 March 2018 from global latitude −10 to 90, The size is 40 Gb, in range 2880 × 561, one elevation segment, and the update frequency is 12 h. The edge computing environment was mimicked by a cluster including three node Kafka cluster, three node zookeeper cluster, and one main node plus eight working nodes. All nodes were configured as four cores, 8 G memory and Gigabit NIC. The experiment tested architecture is shown in Figure 9. Kafka, as a high throughput distributed publish–subscribe message system, was utilized to distribute tasks for the segmented dataflow. Spark [47] was responsible for parallel dataflow processing. HBase was used to implement the stable storage module.To show how our framework advances the MVFD processing, we compared our framework against widely used classical big data processing frameworks, i.e., MapReduce and classical Spark.

In the experiment, we deployed a spark cluster consisting of one driver and six executors. We set the value of executor.core to 2, executor.memory to 4, driver.core to 4, driver.memory to 8 G, and kept other parameters as default values. In MapReduce, the node configuration was the same as that of Spark cluster, setting the value of mapreduce.am.resource.mb to 4 G, app.mapreduce.am.resource.cpu-vcores to 2, mapreduce.map.memory.mb to 4, mapreduce.map.cpu.vcores to 8, and keeping other parameters as default values. In DFS, the configuration for Kafka cluster was: one topic as input resource, including 12 partitions, and replication set to 1. Another topic as result output resource, including eight partitions, and replication set to 1. Others configurations were the same as those for Spark experiments. In this way, the interference of hardware and parameter configuration could be eliminated in experimental comparison.

### 5.2. Experiment Results and Analysis

Let us first look at the total data processing time of the three frameworks on the two datasets. The results are reported in Figure 10. Obviously, we first notice that our algorithm exhibited advantages over the two competitors in both datasets. We attribute such advantage to our careful scheduling of data fluidization, data transmission and data computation. This verified the correctness of our design. Next, let us dive into the details of MVFD processing to give more insightful understanding on why our framework outperformed existing ones.

Data partition is critical in our framework. We first investigated how it affects the total data processing time. We increased the data block size from 0.1 MB to 100 MB and reported the results in Figure 11, from which we noticed that the processing time first decreased, and then increased, with the increasing of data block size. When the size of data block was less than 3 MB, the computational performance increased with the increase of data block. This is because, when the parallelism was saturated, with the increase of data blocks, the communication time decreased. However, when the size of data block was larger than 3 M, the computational performance decreased with the increase of data block. Although the communication time decreased, the total processing time still increased due to the decrease of computing parallelism. This indicates that we must carefully decide the block size to balance the time cost on communication and computation.

Next, we further investigated how the iteration times (*N*) affects the overall performance. Increasing the iteration time from 11 to 111, the evaluation results are shown in Figure 12. Obviously, with the increase in the number of iterations, the processing time increased for all algorithms. Nevertheless, our framework still always reserved the best performance among the three competitors. For the performance analysis of subset computation of two datasets, the data processing time experienced by the customer is shown in Figure 13. The results further verified the high efficiency of our framework by the fact that it always outperformed existing mainstream frameworks in any data block size and any number of iterations. Our proposed framework is suitable for MVFD processing in the consideration of overall processing time.

## 6. Conclusions

MVFD processing is widely used in many application domains, e.g., climate, hydrology, etc. How to improve the performance efficiency of MVFD processing therefore is a significant but challenging problem. Taking the advantages of data source proximity, we leverage edge computing as complementary to cloud computing and propose an edge computing MVFD processing framework, in which data fluidization plays a critical role. To this end, we invent a data fluidization strategy that realizes full utilization of communication and computation resources in the edge computing environment. In addition, it also enables balanced communication and computation scheduling such that the total processing time can be minimized. To verify the efficiency of our proposed framework, we practically implemented it and tested its performance against classical mainstream frameworks, i.e., MapReduce and Spark. The results show that our framework indeed achieved our design goal to reduce the total MVFD processing time and exhibited performance advantages over both MapReduce and Spark.

The framework proposed in this paper is currently based on Spark-streaming and Kafka, which can save the workload of the underlying development to a certain extent. Assuming that there are better flow computing frameworks and message distribution tools than spark-streaming and Kafka in the future, we only need to replace spark-streaming and Kafka on the basis of DFS to achieve high performance calculation of MVFD.

## Figures and Tables

**Figure 1 sensors-19-02602-f001:**
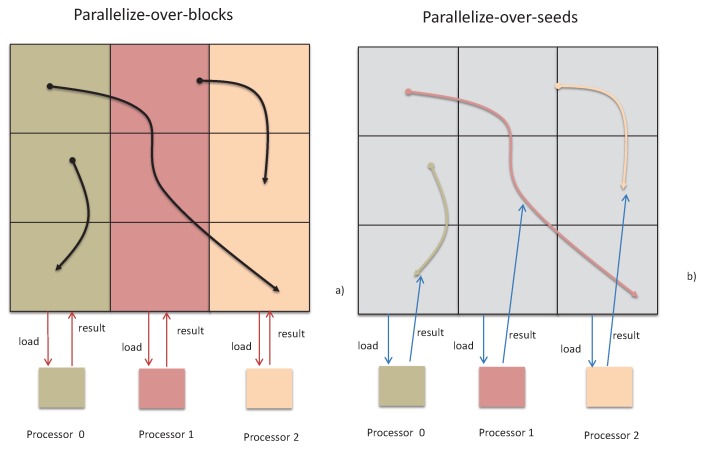
Two typical parallel vector field data processing paradigms. (**a**) In parallelize-over-blocks, each server preloads and calculates part of the data. (**b**) In parallelize-over-seeds, each server preloads all the data, but calculates only part of them.

**Figure 2 sensors-19-02602-f002:**
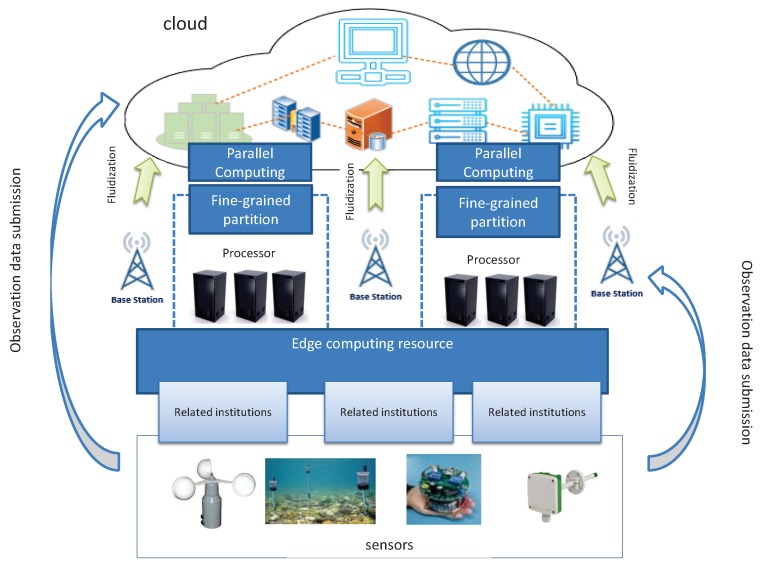
Edge computing empowered MVFD processing framework.

**Figure 3 sensors-19-02602-f003:**
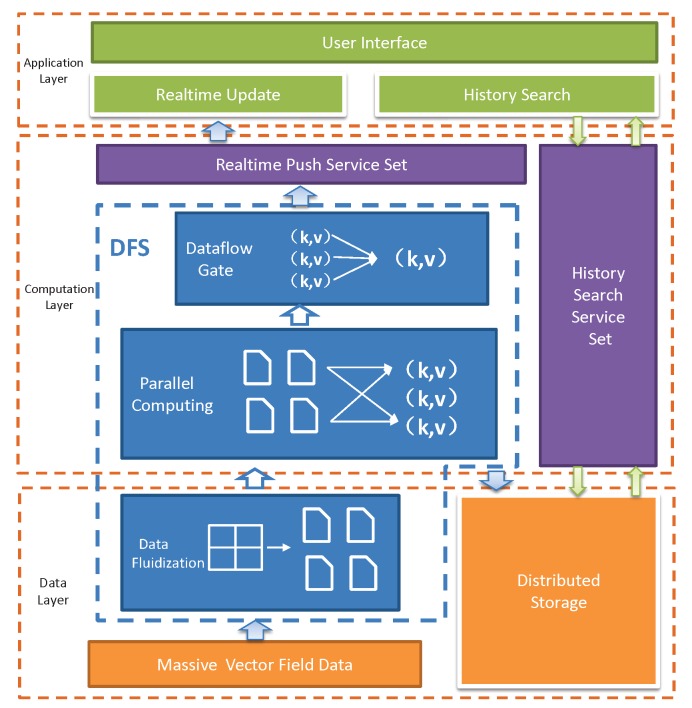
Overview of data process workflow.

**Figure 4 sensors-19-02602-f004:**
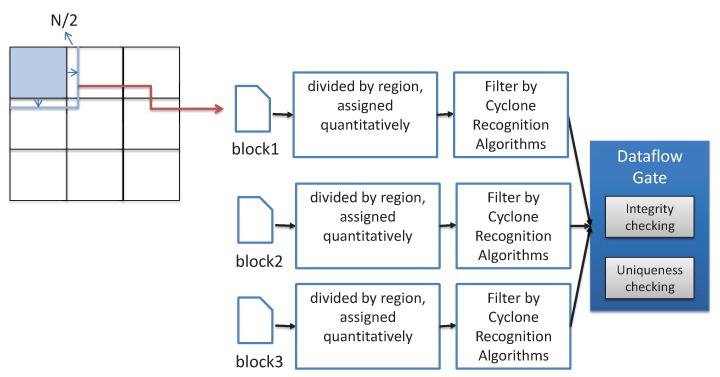
Cyclone recognition process.

**Figure 5 sensors-19-02602-f005:**
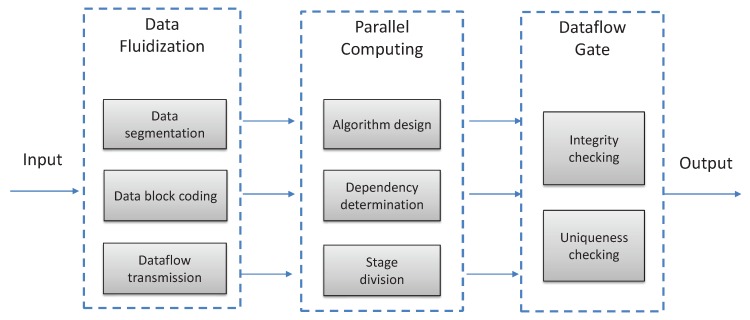
Flow-process diagram of DFS. This pattern contains three parts: data fluidization, parallel computing, and dataflow gate.

**Figure 6 sensors-19-02602-f006:**
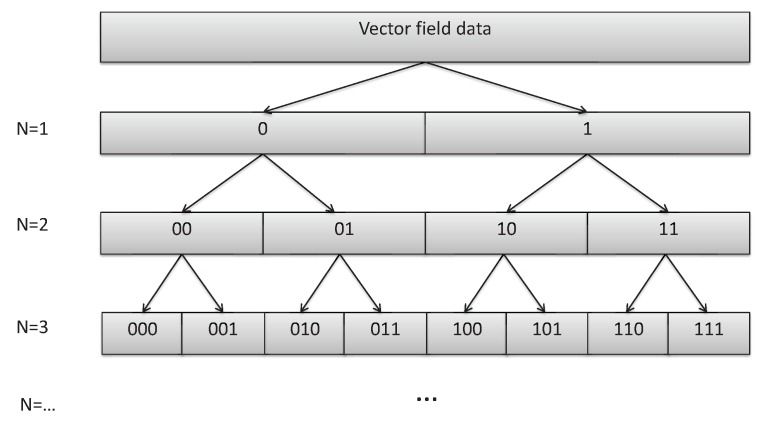
Special segment process. In data fluidization, data are divided and encoded according to spatial hierarchy. Data can be divided according to row, column, quartile, irregular graph and so on.

**Figure 7 sensors-19-02602-f007:**
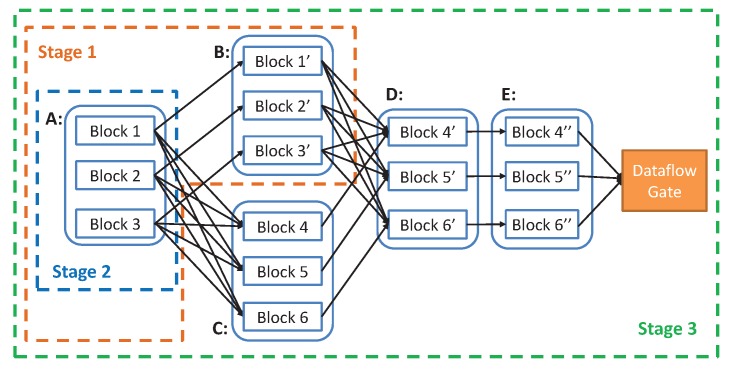
Example of parallel computing on dataflow. In this scenario, parallel computing is divided into three stages (Stages 1, 2, and 3), among which Stage 1 can be computed in the process of data streaming transmission.

**Figure 8 sensors-19-02602-f008:**
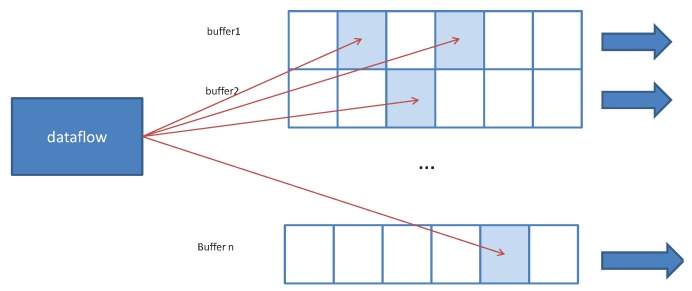
Dataflow gate structure. Buffer stores incomplete results. When the buffer is saturated, it outputs the entire result. When there is data duplication, it produces coverage in the buffer, avoiding duplicated output.

**Figure 9 sensors-19-02602-f009:**
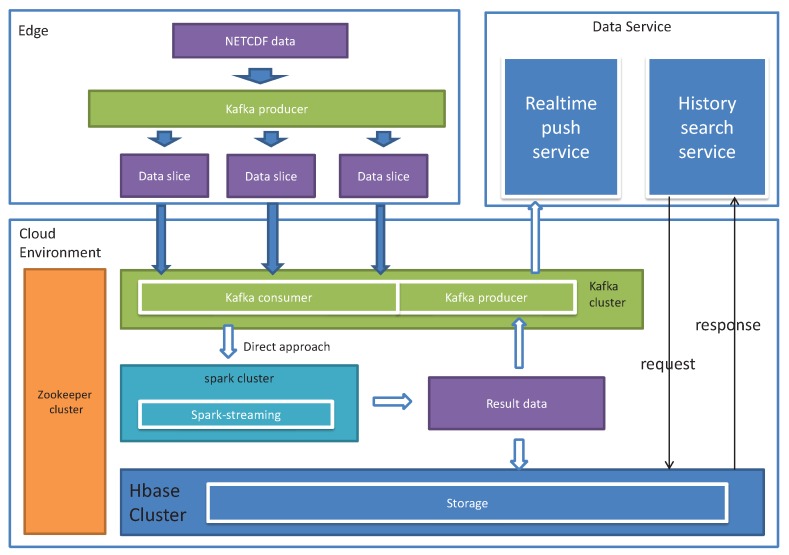
Technical architecture of the application. In the experiment, data fluidization was performed on the producer side of Kafka on the data source, and parallel computation was performed on Spark-Streaming. Dataflow gates were configured on the producer side of Kafka in the cluster environment.

**Figure 10 sensors-19-02602-f010:**
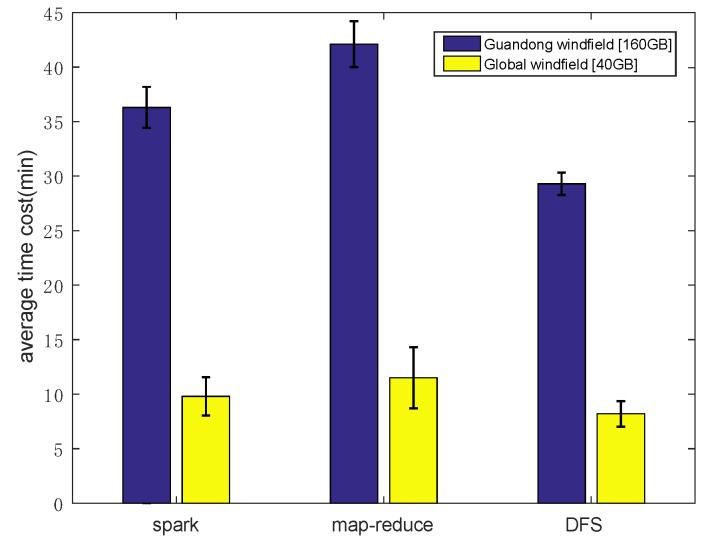
Average total time consumption comparison under different frameworks. The performance of DFS was obviously better than that of mainstream computing framework when calculating massive vector data.

**Figure 11 sensors-19-02602-f011:**
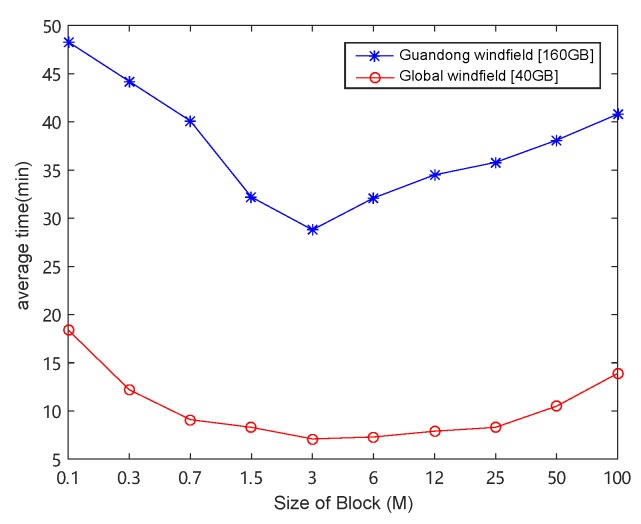
The effect of block size on the overall computing efficiency. It can be seen that when the block size was about 3 M, the frame had better performance. When data were greater than or less than 3 M, the increase or decrease of data led to lower performance.

**Figure 12 sensors-19-02602-f012:**
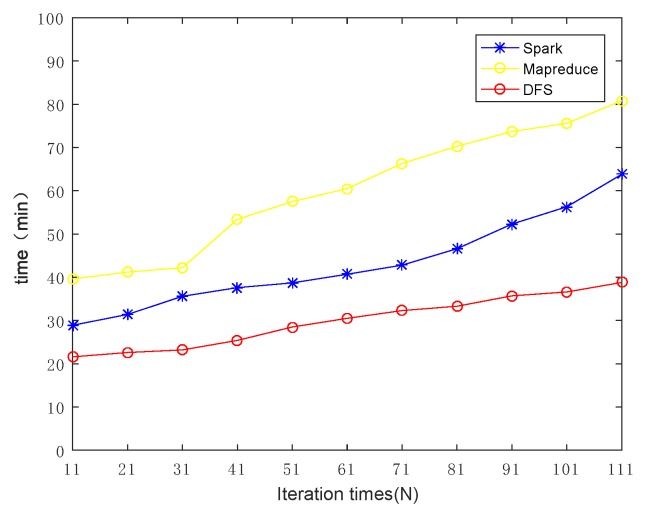
The effect of iteration times on the overall computing efficiency. It can be seen that, with the increase in the number of iterations, the time consumption of DFS was less than the main stream framework.

**Figure 13 sensors-19-02602-f013:**
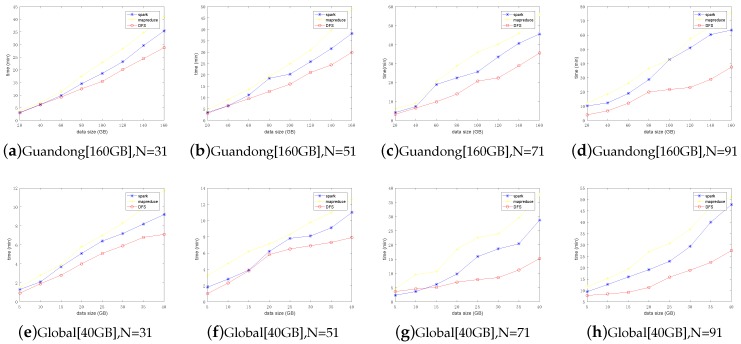
Comparison of timeliness of different frameworks. It can be seen that DFS has better performance advantages than other frameworks when there are more data. With the increase of *N*, the number of calculation cycles increased, and the performance advantage of DFS became more obvious.

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
