# Peer review of "Real-Time Massive Vector Field Data Processing in Edge Computing"

_sensors, 2019, doi:10.3390/s19112602_

Round 1

Reviewer 1 Report

In this paper authors propose a edge computing based vector field data processing architecture, which outperform Apache Spark and MapReduce according to the experiments.

I have the following suggestions and issues

Questions/Suggestions

- It is well-written and understandable paper. Yet, the experiments still depend on Kafka and Spark Streaming. In that sense, it still depends on the possible optimizations and processing advantages provided by these two platforms. I would like to learn about insights of authors about the possibility of not using any such a framework for the implementation and experiments.

- I would like to ask authors to share the data or cite the source of it (papers and URL) that they are used for experiments. This is necessary for the reproducibility of experiments.

- There is no information about how Spark and MapReduce are configured. Did you try to find best configuration parameters (which may severely effect the performance) for the Spark and MapReduce or did you use the default configuration (especially for experiments leading to Figure 10)? Please also share the configuration parameters of these frameworks.

Small issues

- L128-129: What do you mean with "There tasks are generally called as Data Fluidization Schedule (DFS) in MVFD processing, which has deep influence on the overall MVFD processing performance efficiency." sentence? Did you mean "These tasks"?

- Algorithm 2: comsumer -> consumer

- Algorithm 2: while ture -> while true

- L239: motivated use -> motivated to use

- Figure 9.: Application technical route -> Technical architecture of the application

- Mapreduce/mapreduce -> MapReduce

- spark -> Spark

Author Response

Dear reviewer,

We have revised our paper in accordance with your comments and marked it in red font. Specific point-by-point modifications are described in the following annexes.

Reviewer 2 Report

This solution proposal paper presents an edge computing based massive vector field data (MVFD) processing framework.  MVFD is generated by vastly distributed sensors and is characterized by high distribution, high velocity, and high volume. As a result, computing such kind of data on centralized cloud infrastructures faces several challenges like the processing delay due to the distance between the data source and the cloud. Therefore, the authors taking advantage of data source proximity of edge computing to solve this problem.

The main contribution of the paper is the proposal and evaluation of a Data Fluidization Schedule (DFS) algorithm that reduces the data block volume and I/O latency. The evaluation is done by a practical application

on massive wind field data processing for cyclone recognition. It is compared with two common big data processing frameworks Spark and MapReduce.

The paper is well written, concise and fits the intended purpose and scope of the journal.

There is little to criticize about the paper and the proposal. However, the authors could work on the following points to even improve the overall quality of the paper.

- Line 33: Instead of "huge batch" the term "high volume" should be prefered

- The introduction could reflect a bit more on the differences in cloud computing and edge computing. Maybe this reference https://www.researchgate.net/publication/312045183 would be suitable to address the cloud computing aspect a bit more?

- Algorithm 2 has several typos: Comsumer -> Consumer; Ture -> True; And because of the while true loop, the algorithm would never terminate. However, I do not think that the authors intended to express that. In line 12, I assume that the value of res should be sent asynchronously at that point. However, this should be expressed somehow differently. It can not be expressed in a pure sequential chain of statements.

- General remark for Section 5: The evaluation compares DFS, Spark, and MapReduce. However, the specific experiment setting is not explained for each of these three experiments. These three experiment settings should be added. Furthermore, it should be made manifest, that all three experiments had been run on an identical hardware-setting. So, the only variable changed should be the compared frameworks (DFS, Spark, MapReduce). However, this is hardly derivable from Figure 9.

- Figure 10: The dataset labels should be more descriptive than Dataset1/Dataset2 (e.g. Guandong windfield [160GB], Global windfield [40GB])

- Figure 11: y-label (average time costs -> average time) Why costs? More descriptive dataset labels (see my remark for Figure 10).

- Some grammar and typos are still in the submission. The authors may let an English native speaker do the final proof-reading or should make use of Grammar checkers like Grammarly (premium edition).

Author Response

(The authors gave the same response as above.)
